# Multifunctional 3D-Printed Magnetic Polycaprolactone/Hydroxyapatite Scaffolds for Bone Tissue Engineering

**DOI:** 10.3390/polym13213825

**Published:** 2021-11-05

**Authors:** Mauro Petretta, Alessandro Gambardella, Giovanna Desando, Carola Cavallo, Isabella Bartolotti, Tatiana Shelyakova, Vitaly Goranov, Marco Brucale, Valentin Alek Dediu, Milena Fini, Brunella Grigolo

**Affiliations:** 1REGENHU Ltd., Z.I. Le Vivier 22, 1690 Villaz-St-Pierre, Switzerland; mauro.petretta@ior.it; 2SSD Laboratorio RAMSES, IRCCS Istituto Ortopedico Rizzoli, 40136 Bologna, Italy; isabella.bartolotti@ior.it (I.B.); brunella.grigolo@ior.it (B.G.); 3SC Scienze e Tecnologie Chirurgiche, IRCCS Istituto Ortopedico Rizzoli, 40136 Bologna, Italy; alessandro.gambardella@ior.it (A.G.); milena.fini@ior.it (M.F.); 4Istituto per lo Studio dei Materiali Nanostrutturati, Consiglio Nazionale delle Ricerche, 40129 Bologna, Italy; biodevicesystems@gmail.com (V.G.); marco.brucale@cnr.it (M.B.); valentin.dediu@cnr.it (V.A.D.); 5BioDevice Systems, Bulharská, 10-Vršovice, 996/20, 10100 Praha, Czech Republic

**Keywords:** polycaprolactone/hydroxyapatite scaffolds, 3D additive manufacturing, superparamagnetic nanoparticles, nanocomposites, tissue engineering

## Abstract

Multifunctional and resistant 3D structures represent a great promise and a great challenge in bone tissue engineering. This study addresses this problem by employing polycaprolactone (PCL)-based scaffolds added with hydroxyapatite (HAp) and superparamagnetic iron oxide nanoparticles (SPION), able to drive on demand the necessary cells and other bioagents for a high healing efficiency. PCL-HAp-SPION scaffolds with different concentrations of the superparamagnetic component were developed through the 3D-printing technology and the specific topographical features were detected by Atomic Force and Magnetic Force Microscopy (AFM-MFM). AFM-MFM measurements confirmed a homogenous distribution of HAp and SPION throughout the surface. The magnetically assisted seeding of cells in the scaffold resulted most efficient for the 1% SPION concentration, providing good cell entrapment and adhesion rates. Mesenchymal Stromal Cells (MSCs) seeded onto PCL-HAp-1% SPION showed a good cell proliferation and intrinsic osteogenic potential, indicating no toxic effects of the employed scaffold materials. The performed characterizations and the collected set of data point on the inherent osteogenic potential of the newly developed PCL-HAp-1% SPION scaffolds, endorsing them towards next steps of in vitro and in vivo studies and validations.

## 1. Introduction

Critical-sized bone defects (CBDs) represent a serious health problem worldwide and remain one of the leading clinical orthopedic challenges among scientists [1,2,3,4,5]. Besides its structural properties, bone displays relevant biological functions, including calcium and phosphate storage and bone marrow harboring. Although bone tissue presents a well-known remodeling capacity, large CBDs do no heal spontaneously [6,7]. The gold standard treatment for bone lesions is envisage graft, taken from the patient itself or multiorgan donors [8]; however, this approach shows several limitations, including the low availability of implant material, donor site morbidity and the potential risk of immunogenic responses [9,10]. Recently, bone tissue engineering (BTE) has implemented alternative strategies to face this unmet clinical need.

BTE aims to develop biological substitutes that restore, maintain, or improve tissue functions by combining biomaterials, cells and bioactive molecules [11]. Mesenchymal stromal cells (MSCs) can provide significant benefits in promoting bone regeneration because of their differentiation and paracrine properties [12]. Biomaterials play a core role in maintaining cell functions with huge implications for the success of the implants. A scaffold has to match the proper structure and suitable mechanical properties to support biological processes and withstand mechanical loading to promote bone regeneration [13,14,15,16].

To this end, researchers have considered a wide range of biomaterials, including natural and synthetic polymers, ceramic, and metals [15,16,17,18,19]. Calcium phosphates, such as hydroxyapatite (HAp) [20,21], tricalcium phosphate [22,23] and bioglasses [24,25] are among the most used and promising ceramic materials, able to promote mineralization processes due to their natural properties [26,27,28,29]. These ceramic materials are often doped by various ions in order to improve their mechanical, bioactive and degradation behavior [30,31,32]. However, pure ceramic scaffolds cannot support cyclic loading due to their brittle behavior, which cannot ensure proper mechanical loading, thus being able to lead to implant failure [33]. Technological advances allowed to improve the structural features of scaffolds for bone regeneration through the incorporation of calcium phosphates as micro-nanoparticle fillers within different polymeric matrices, such as collagen [34,35], gelatin [36,37], silk fibroin [38,39], chitosan [24,40], poly(propylene fumarate) (PPF) [41], resorbable polymers [42,43] and others. Among polymers, polycaprolactone (PCL) is one of the most used candidates as a matrix for bone regeneration. PCL is an FDA approved thermoplastic polymer, it is easily processable (low melting temperature, high stability), biocompatible and biodegradable. PCL displays degradation times tunable through the choice of adequate molecular weight, and features suitable mechanical properties for BTE, especially when combined with reinforcing fillers [44,45,46,47,48,49]. Regarding the structural characteristics of biomaterials, the control over internal scaffold architecture is another critical parameter in BTE. To this end, multiple approaches have been used for BTE [4,9,10,16,50], including Additive Manufacturing (AM) and rapid prototyping techniques. AM techniques allowed designing structures with controlled internal architecture at the microscale level (fiber and pore size, biomaterial and bioactive deposition, and cell placement) by computer-Aided Design and Manufacturing (CAD/CAM). Moreover, this technique allows fabricating patient-specific and personalized constructs through customized models based on imaging techniques such as computer tomography (CT) and magnetic resonance imaging (MRI) with enormous perspectives in the medical field [51,52,53,54,55].

In addition to the proper choice of scaffold materials and structure, it is important to consider that the implantation of cell-free scaffolds has limited healing effects and it is often aided by the in vitro pre-loading of scaffolds with necessary bio-agents (cells, growth factors etc.) before the implantation [56]. Despite various employed efforts and many promising results, clear and useful protocols for an efficient and homogeneous scaffold cellularization are still missing [57].

The employment of magnetic nanoparticles in biomedical applications have attracted a considerable interest as labels or for various manipulations with biomaterials and tissues both in vitro and in vivo [58,59]. The magnetic labeling allows for enhanced detection of various bioagents [60] and enables the high-resolution magnetic particle imaging [61]. Moreover, in vivo hyperthermia treatment can be pursued upon the labeling of tumor tissues with magnetic nanoparticles [62]. Considering the magnetic manipulations, the application of magnetic nanoparticles is mainly employed to transport and extract biomaterials, and to deliver required drugs to target tissues [59]. The preference in magnetic manipulations is given to superparamagnetic nanoparticles typically represented by iron oxide (SPIONs), which allow for an efficient manipulation, combining high magnetization in external magnetic fields with no magnetic remanence in zero field, the latter excluding or significantly reducing the aggregation of nanoparticles in random-sized objects [57,63,64,65]. More recently, magnetic scaffolds have been additionally introduced to achieve locally strongly concentrated magnetic fields and gradients, enhancing further the efficiency of magnetic treatments for tissue engineering [57].

The study presented here aims at a straightforward demonstration of the advantage of the magnetic functionality embodied in scaffolds for an efficient cell seeding. To achieve this, we developed novel PCL-based scaffolds containing both HAp and SPIONs to enhance the efficiency and the control of the cell homing. The bulk distribution of HAp and nanomagnetic species was extrapolated from the surface distributions detected by atomic force and magnetic force microscopies (AFM-MFM). The magnetic seeding of cells inside the scaffolds was studied by using NIH/3T3 fibroblast line, while viability assay and osteogenic potential were tested on MSCs seeded onto a PCL-HAp-1% SPION scaffold by standard procedures.

## 2. Materials and Methods

### 2.1. Magnetic Nanocomposite Preparation

The functionalized composite formulation was obtained by using PCL as a matrix and SPION nanoparticles (average diameter 50–100 nm) (Sigma, St. Louis, MO, USA) and HAp microparticles (average diameter below 10 µm; Bioteck, Vicenza, Italia) as fillers. Two different concentrations of SPION, 0.5 and 1% were selected, while HAp was included as 10% of the total weight. This concentration for Hap was kept in order to guarantee an improved biological response while at the same time providing an effective mechanical reinforcement, as reported by previous studies in the field [64,66]. To select the optimal magnetic fraction, we moved by increasing the concentration from low to high, trying to accomplish a compromise between as low as possible concentration of magnetic material while achieving efficient cell seeding. Three different formulations of scaffolds were developed, identified as follows: non-magnetized PCL with 10% HAp (PCL-HAp), magnetized PCL with 10% HAp and 0.5% SPION (PCL-HAp-0.5% SPION) and magnetized PCL with 10% HAp and 1% SPION (PCL-HAp-1%SPION). For the realization of the composites, PCL pellets with a molecular weight of 80,000 (Sigma Aldrich, St. Louis, MO, USA) were dissolved in Chloroform. The required amount of HAp microparticles and nanometric SPION to reach the desired concentration was then gradually added to the solution under magnetic stirring. The final solutions were left under stirring overnight at room temperature (RT) to guarantee a proper mixing. To further improve the particle dispersion homogeneity and avoid clustering, the composite solution was further sonicated for 30 min before precipitation. Finally, composite pastes were obtained by precipitating in excess of absolute ethanol. The obtained paste was kept under the chemical hood for 24 h to remove residual solvent before being pelletized.

### 2.2. Scaffold Design and Fabrication

Magnetic nanocomposite scaffolds were fabricated as follows. First, cubic (5 mm size) structures were designed through BioCAD software (RegenHU, Villaz-St-Pierre, Switzerland). The fiber size was set to 330 µm and the distance between two adjacent fibers to 250 µm. As for the deposition pattern, a 0/90° pattern with an offset of 290 µm, equal to half inter-filament size (distance between adjacent fiber centers) between two following repeating units was chosen (0/90/0_off_/90_off_). The layer height was set to 250 µm (approximately 80% of the fiber diameter) to guarantee an adequate fiber stacking. The printing process was performed through the screw-based extruder printhead of a 3D Discovery platform (RegenHU, Villaz-St- Pierre, Switzerland). The previously obtained composite pellets were loaded in the printhead tank, which was kept at 70 °C during the process. After 30 min, the printing process was started. The pressure was set to 3 Bar, the barrel temperature to 90 °C and the screw rotation speed to 20 rpm. A deposition speed of 12 mm/s was chosen for the printing process. Construct sterility was ensured by performing the process in a Class II biosafety cabinet (where the 3D Discovery platform is embedded). We replicated the printing in an automated manner, using the BioCAD software, directly within 6-well low attachment culture plates (Corning Inc., New York, NY, USA). A total of 30 cubic scaffolds were fabricated for each formulation for the characterization tests. Single filaments were extruded with the same printing parameters and conditions and subsequently collected for the magnetic force microscopy characterization.

Scaffold porosity was calculated from the CAD design through the modeling approach optimized in a previous study from our group and described in [47], which provided a final porosity value of 45.81%.

### 2.3. Magnetic Seeding of Cells

#### 2.3.1. Cell Magnetization

The capability of seeding the magnetic scaffolds with cells was tested by using NIH/3T3 fibroblast line (ATCC^®^, CRL-1658^TM^). Note that the magnetic seeding of MSC cells, used further in this paper, in similar scaffolds has been successfully accomplished previously by part of the authors and reported in a current publication [57]. Fibroblasts at a concentration of 3.5 × 10^4^ cells/cm^2^ were seeded in T75 flasks and incubated with α-MEM growth medium (Sigma, St. Louis, MO, USA), supplemented with 10% of fetal bovine serum (FBS) (Sigma, St. Louis, MO, USA), 1% HEPES (Sigma, St. Louis, MO, USA), 1% sodium pyruvate (Sigma, St. Louis, MO, USA), and 1% penicillin-streptomycin solution (Sigma, St. Louis, MO, USA). The cells were left to adhere to bottom of the flask at 37 °C in 5% of CO2. Once the cells adhered and achieved a sub-confluent monolayer, 5 μL of ferrofluid based on 10 nm large SPION was added to the culture medium, and the cells were incubated overnight. After 24 h, the medium was changed and replaced with fresh medium. When the cells reached confluence, they were trypsinized and centrifuged at 1000 rpm for 5 min at RT. Finally, the cells were diluted in 10 mL of the afore-mentioned medium at 10^5^ cells/mL concentration and used for scaffold seeding. The level of cell saturation with nanoparticles achieved by this procedure was approximately 100 pg of SPION per cell, a concentration, sufficient for the magnetic manipulation of cells by standard permanent magnets [57,66].

#### 2.3.2. Experimental Setup for Scaffold Seeding

The scaffolds were characterized by three different levels of magnetization reported above: high magnetization (PCL-HAp-1%SPION), low magnetization (PCL-HAp-0.5%SPION) and non-magnetized scaffolds (PCL-HAp). The seeding of cells was performed following two different schemes. In the first, we set single scaffolds between two NdFeB permanent magnets (diameter 15 mm, length 5 mm, Br = 1.2 T) distanced 27 mm and performed the seeding of cells for the three types of scaffolds by perfusing them with the 10^5^ cells/mL concentration (described above). The second scheme employs a more complex approach, by placing thirteen scaffolds of different magnetizations in an array inside the flow tube (Figure 1) and perfusing them simultaneously. The scaffold N9 from the second scheme is set with respect to magnets in the same position as the single scaffolds in the first scheme. In both configurations, the scaffolds were perfused by magnetized cells in an in-house developed flow bioreactor for 3 h at 1 mL/min flow velocity. For better cell seeding uniformity, especially for scaffolds array, the flow direction was inverted every 20 min. After seeding, the concentration of viable cells has been evaluated using standard resazurin reduction test (Sigma, St. Louis, MO, USA).

### 2.4. Magnetic Simulations

Computer modelling with COMSOL3.5 (Comsol Inc., Stockholm, Sweden) was employed to calculate the radial distribution of magnetic field in the middle of two cylindrical permanent magnets using Multiphysics–Magnetostatics–2D axial symmetry application mode.

### 2.5. Atomic Force Microscopy-Magnetic Force Microscopy (AFM-MFM)

Scaffold morphology was evaluated by Bruker Multimode8 AFM (Bruker Nano Surface, Milan, Italy) equipped with a Nanoscope V controller, a JV type scanner and operating in semi-contact mode at room temperature. The instrument implements Magnetic Force Microscopy (MFM) and uses magnetic Ultrasharp NSC36/Ti–Pt probes (MikroMasch, Sofia, Bulgaria). The microscope was operated in double-pass Lift Mode with lift heights ranging from 30 to 500 nm and a tip bias of ± (100 ÷ 500) mV. The MFM images (both topographies and phase signals) were acquired at 256 × 256 pixel resolution (Appendix A).

### 2.6. Cell Isolation, Characterization and Expansion

MSCs, obtained from the bone marrow of an anonymous human donor, were selected from a pool stored in the Biobank of our lab at the Rizzoli Orthopedic Institute. Experiments were performed in triplicates. Cells were expanded for two passages and characterized by FACS analysis for the presence of typical mesenchymal and hematopoietic markers and their clonogenicity ability [67,68].

### 2.7. MSC Seeding and Viability

Two different kinds of pre-treatments, 10% FBS and polylysine (Sigma-Aldrich; Burlington, MA, USA), were tested to enhance MSCs attachment on PCL-HAp and PCL-HAp-SPION scaffolds. After 30 min of incubation, 1 × 10^6^ MSCs resuspended in 75 µL of culture medium per scaffold were placed in ultra-low attachment 6-well plates (Corning Inc., New York, NY, USA). Inverted light microscopy and viability analyses were used in the first instance to test the MSCs behavior on both scaffolds. Cell survival and proliferation were tested by Alamar-blue assay according to the manufacturer’s instructions.

### 2.8. Osteogenic Differentiation of MSCs-Laden 3D Scaffolds

Research design foresaw the culture of 1 × 10^6^ MSCs for each scaffold and assessment of their osteogenic potential in PCL-HAp and PCL-HAp-SPION structures with and without growth factors (GF) at 1, 14 and 21 days. At culture passage 2, we previously pre-treated 3D PCL-HAp and PCL-HAp-1% SPION structures with 10% FBS and then performed cell seeding. Constructs were cultured both in an osteogenic medium composed of α-MEM containing 15% FBS, 10^−7^ M dexamethasone (Sigma), 75 µg/mL ascorbate-2 phosphate (Sigma, St. Louis, MO, USA), and 0.01 mM β-glycerolphosphate (Sigma, St. Louis, MO, USA) and in α-MEM alone. 2D cultures of MSCs were used as controls.

### 2.9. Histological Analyses

Alizarin Red staining was performed on both 2D and 3D cell cultures to evaluate the synthesis of the mineralized matrix [56]. We fixed 2D cultures with buffered formalin (Kaltek S.r.l., Padova, Italy) for 30 min at RT and stained with the Alizarin Red S solution (Sigma Aldrich) at days 1, 14 and 21. After the fixation of MSCs laden-PCL-HAp and PCL-HAp-1% SPION, we performed the dehydration in a graded series of alcohol and K-clear plus, a xylene substitute (Kaltek) before polyester wax embedding (Kaltek). We cut serial sections (thickness: 10 µm) with a rotary microtome followed by Alizarin Red S staining. We assessed the neo-formed mineralized matrix using ImageJ software (NIH, Bethesda, MA, USA) by defining a threshold with the red, green blue (RGB) system. As for cell distribution, we performed the staining of sections from the upper, mid and bottom parts of scaffolds with Hematoxylin/Eosin (BSPIONtica, Milano, Italy). We incubated samples with Gill III Hematoxylin for 30 s, followed by the activation with tap water for ten minutes and then the staining with Eosin for 3 min at RT.

### 2.10. Statistical Analysis

GraphPad Prism 7 software (San Diego, CA, USA) was used for statistical analysis. Normal distribution of data was evaluated using the Kolmogorov-Smirnov test. Kruskall-Wallis with the post hoc Dunn’s multiple comparison test allowed assessing differences in cell viability and osteogenic potential in MSCs laden-PCL-HAp and PCL-HAp-1% SPION at days 1, 14 and 21. Data are reported as mean ± standard deviation. Data reporting *p* < 0.05 were considered significant. Technical triplicates were used for all the experiments.

## 3. Results

### 3.1. Magnetic Seeding of Scaffolds

The magnetic seeding of scaffolds with cell solutions described in Section 2.3.1 was performed in two steps. In the first step we investigated the seeding of single scaffolds with three magnetization levels described above, repeating the experiment three times for each magnetization. After the accomplishment of the seeding process, we registered the following numbers of cells: (1.4 ± 0.09) × 10^4^ in PCL-HAp-1%SPION scaffolds; (1.2 ± 0.1) × 10^4^ in PCL-HAp-0.5%SPION scaffolds and (1.0 ± 0.1) × 10^4^ in PCL-HAp scaffolds (Figure 2a) achieving thus a higher cell concentration inside the scaffolds with higher magnetization.

We further employed a more complex scheme perfusing simultaneously thirteen mixed scaffolds in the presence of magnetic field (Figure 2b). The seeding of scaffolds by cells followed the same procedure as for single scaffolds. It is important to note, that independently of the scaffold magnetization, the designed geometry and the employed loading scheme of scaffolds immersed in cell medium and flowing the medium back and forth, leads to a non-uniform distribution of seeding efficiency with a higher cell concentration in the edge scaffolds and minimal cell concentration in the central ones. This is due to the gradual reduction of the concentration of cells absorbed first by edge scaffolds. This overall trend was experimentally confirmed and the polynomial fitting in Figure 2c shows that the scaffolds are regularly scattered around the curve, with seeding cell amounts ranging from 0.23 × 10^4^ cells for edge scaffolds to 0.10 × 10^4^ cells for central scaffold N6. Nevertheless, a clear and indicative deviation from this trend was detected on the scaffolds except the N8, N9 and N10. The three specified above scaffolds were placed in the region of the highest magnetic intensity highlighted in Figure 2b–d and behaved distinctly different, showing a cell seeding efficiency clearly above the average trend.

Figure 2d shows a quantitative analysis of this specific seeding distribution through the deviation values from the fitting line, the latter calculated without considering the magnets N8–N10. The deviation does not exceed the 0.02 × 10^4^ value, except the three scaffolds set between the permanent magnets. In addition, note the difference between the seeding efficiency of the two black strongly magnetized scaffolds (N8 and N10) and the non-magnetic white scaffold N9. Summarizing, the addition of SPIONs to the PCL-Hap scaffolds provides a higher cell seeding efficiency, which nevertheless is activated only upon the application of the external magnetic field of sufficient intensity (20 mT in our case).

### 3.2. AFM-MFM Measurements

MFM is a high-spatial-resolution technique to image magnetic properties at nano- and microscale. It provides information about stray field distribution above the sample that is connected with sample magnetization and uses the magnetic forces (or the force gradient) acting between the magnetized sample surface and the magnetized tip [69]. The measurement typically consists in determining the change in phase of an oscillating cantilever on which the tip is mounted. The phase changes according to whether an attractive or repulsive force exists between the local magnetization of the sample and the moment of the tip. In this context, the exploitation of this technique allows in principle to detect the weak magnetic signals within the SPION-charged scaffolds. We underline that even though the bulk of the scaffold cannot be visualized by these methods, characterizing the surface is of primary importance due to its stronger interaction with the environment. As concerns the topography of the fiber surface, shown in Figure 3a, the texture consists in rod-like segments which are typical of polymer [70], together with ~µm in size protrusions or smaller grains randomly distributed over the surface. Similar topographic features characterize the surface among all three scaffold formulations; moreover, the detection of magnetic phase signal allows to discriminate between topographic and magnetic contrast in Figure 3. Nevertheless, detection of magnetic contrast in the PCL-HAp-0.5% SPION formulation by MFM was unclear due to the low variation in the phase signal (less than a few degrees), whereas the PCL-HAp-1%SPION one exhibited phase variations of several degrees, lending itself to qualitative evaluation. In Figure 3a, the topographic feature A (~4 µm large), in correspondence of which no relevant contrast in the magnetic phase image was detected, may be ascribed to a HAp particle which protrudes of about 250 nm from the fiber surface. The elongated feature B has instead very little topographic contrast but remarkable magnetic contrast; thus, it is likely the contour of a magnetic domain slightly protruding from the surface. These findings could be reasonably ascribed to an on-surface description of the internal structure and distribution of the two components inside the fiber.

### 3.3. Biological Features of MSCs

MSCs surface markers were analyzed by flow cytometry. Cells were positive for typical mesenchymal markers like CD-146, -106, -105, -90, -73, -44 and negative for typical hematopoietic markers like CD-45, -34, -31, -14. The clonogenicity capacity was also evaluated, MSCs formed colonies at both 10 and 20 days (Appendix A).

### 3.4. Cell Distribution and Viability Assessment of MSCs Seeded onto 3D Printed Constructs

Both pre-treatments with 10% FBS and poly-lysine displayed similar results in terms of cell distribution and survival. We selected 10% FBS treatment as it is a supplement needed for culturing cells. Analysis with Hematoxylin/Eosin staining showed a regular distribution of MSCs in the upper, mid and bottom of both 3D printed PCL-HAp and PCL-HAp-1% SPION structures. Constructs resulted stable by preserving their architecture with culture media up to day 21 (Figure 4a). Alamar blue assay showed that MSCs seeded onto PCL-HAp and PCL-HAp-1%SPION were metabolically active from day 1 to day 21. Both constructs showed the highest cell proliferation at day 21. MSCs-laden PCL-HAp-1% SPION at day 21 displayed more increased cell proliferation than MSCs seeded onto PCL-HAp at day 1 (*p* < 0.01) (Figure 4b).

### 3.5. Both 3D Constructs Promoted the Osteogenic Differentiation of MSCs

2D cultures of MSCs in basal media without growth factors (GF) showed lower osteogenic potential than MSCs grown in the osteogenic differentiation media with GF. Notably, MSCs cultured with GF displayed a higher osteogenic capacity at day 21 when compared to constructs in the culture medium alone at day 1 (*p* < 0.01) (Appendix A). To test the mineralization ability of MSCs seeded onto PCL-HAp and PCL-HAp-1%SPION under different culture conditions, we performed histological analyses. Frontal histological sections showed a uniform distribution of MSCs in both experimental groups through all the thickness of scaffolds (Figure 5). The PCL-HAp group grown in culture medium alone showed lower mineralization at day 14 than PCL-HAp-1%SPION group treated with GF on day 21 (*p* < 0.01) (Figure 6). Both groups exhibited reduced mineralized areas on day 14; whereas the PCL-HAp-1% SPION group showed the highest presence on day 21 (Figure 6).

## 4. Discussion

Developing scaffolds with improved bone biocompatibility is demanding for several musculoskeletal disorders, represented by the lack or loss of tissue [5]. A better comprehension of bone biology and physiology is essential to design biomimetic scaffolds with suitable biochemical and biophysical cues suitable for bone tissue regeneration. Recently, much effort has been addressed to modulate cell behavior by regulating the biophysical properties of scaffolds like their topographical features (porosity, roughness, patterns) [13,14,15,16]. Additive manufacturing techniques, like stereolithography, selective laser sintering and fused deposition modeling, can allow fabricating customized scaffolds to improve osteoconductivity and osteoinductivity for bone tissue repair [71,72,73]. Notably, the prospect of integrating various inorganic constituents like calcium phosphates within polymer-carriers has the great advantage of designing customized microenvironments [74]. Despite these technological advances, one of the main drawbacks in developing bone scaffolds is due to the limited control of cell function and localization in the defect area [75]. Notably, the use of magnetic scaffolds through the incorporation of iron oxide nanoparticles like magnetite SPION would seem to be a useful tool to drive cells towards targeted tissues via the application of an external magnetic field; thereby increasing tracking duration and reducing non-specific interactions [76]. The benefit of coating SPION on scaffolds is due to preventing the aggregation of extracellular nanoparticles responsible for toxic reactions [77].

In this study, magnetic scaffolds were developed through a screw-based extrusion 3D printing process and then evaluated for their intrinsic osteoinductive potential on MSCs. High-density PCL was selected as the polymer matrix because it owns good biocompatibility, mechanical properties and processability for melt extrusion-based printing technologies [45,78]. Using an automatized 3D printing platform, it was also possible to perform the fabrication process directly within six-well plates in a sterile environment. This approach provided clear and undoubted advantages in terms of repeatability, standardization and sterility of the manufacturing process while increasing user-friendliness at the same time. The study design foresaw the combination of PCL with two different micro/nanofillers, HAp and SPION. This combination provided a suitable microenvironment to foster cell homing and biological responses to the magnetic field. Regarding scaffold architecture, an offset in the layer pattern sequence was selected to ensure an increased surface for cell deposition along their seeding path. The characteristics of PCL-HAp with and without SPION were compared through several analyses to test the impact of SPION on fibroblast behavior. In vitro studies were carried out using a flow bioreactor system with NIH/3T3 fibroblasts, previously magnetized with two different SPION concentrations, for reproducing the magnetic guidance and defining the optimal SPION concentration. The high-magnetization formulation with 1% SPION provided the best results in terms of cell entrapment time and adhesion rates. After having chosen the best SPION concentration, analyses were directed to understanding the biological interactions of SPION with MSCs. Both PCL-HAp and PCL-HAp-1% SPION groups reported an increased cell proliferation up to 21 days; thus, showing that the functionalization with SPION does not affect cell metabolism. These results agree with the literature which reported the benefits of magnetic scaffolds in improving cell proliferation and bone formation [76]. Similar levels of mineralization were found in MSCs seeded on both PCL-HAp and PCL-HAp-1% SPION structures. These findings would suggest that SPION incorporation did not affect the MSCs’ mineralization ability.

The use of the screw-assisted printing technology likely provided benefits in homogenously distributing SPION in the PCL-HAp. AFM-MFM measurements confirmed this homogenous distribution, highlighting intermixed HAp and SPION throughout the surface. Thanks to the high sensitivity and spatial resolution of AFM-MFM techniques, we obtained indications of the small amounts of the scaffolds magnetic components by reducing the risk of damaging their magnetic coating. Detecting the stray field at the nanometer scale in the PCL-HAp-1% SPION gave evidence that scanning probe techniques could be successfully implemented to detect small amounts of the scaffolds’ magnetic component; on the other hand, scanning probe techniques alternative to MFM may be exploited when the amount of magnetization falls below the instrumental limit and a merely qualitative detection of metallic or semi-metallic phase at the surface is sufficient [79].

Previous preclinical studies tested the biological processes underpinning SPION incorporation within several scaffolds by highlighting different proteins related to integrin signaling pathway, MAP/ERK cascades and calcium ions [80,81]. We can speculate that the developed magnetic structures could act similarly through the transduction of “outside-in signaling” towards osteogenic differentiation. The lack of in-depth analyses of the mechanism of action underlying osteogenic differentiation of MSCs-laden PCL-HAp-1% SPION and preclinical in vivo studies are the main limitations of this research. This study showed that both scaffolds provided environmental, physical cues to support the osteogenic differentiation of MSCs also in the lack of specific growth factors in the culture medium. The intrinsic osteogenic potential of these constructs is a crucial determinant to hypothesize future clinical applications avoiding the use of exogenous growth factors by overcoming several issues (definition of concentration, limitations of clinical-grade products, etc.). The magnetic guidance of these constructs within the CBDs could not only contribute to promoting osteogenesis of the resident MSCs but also create a synergic effect on the surrounding endogenous cells within the defect area to secrete growth factors useful for promoting bone regeneration.

Finally, the enhanced seeding of magnetized fibroblasts in magnetic scaffolds reported above, can be understood in terms of currently reported results obtained for similar magnetic scaffolds [57]. It has been shown that a kind of double magnetic effect must be considered for the interaction between magnetized cells and magnetic scaffolds. The first effect comes from the magnetic field itself through the localization of magnetized cells in the area of highest magnetic fields, which increases the cell entrapment time and leads to higher adhesion rates. The second effect comes from the magnetization of scaffolds, and it is based on the short-range magnetic attraction between scaffold fibers and magnetic cells. This force extends to only a few tens of microns, which represents nevertheless the typical cell size and offers additional pinning of cells, further increasing the seeding efficiency. The assessment of this condition requires exploitation of techniques able to distinguish the low amount of magnetization at the local scale, such as AFM-MFM techniques. These seeding experiments confirm the utility of magnetic cellular saturation of scaffolds, able to seed the cells all inside the scaffold body, including its deep parts.

According to the routine protocols of biocompatibility (Practical Guide to ISO 10993-5: Cytotoxicity) the first step to reveal any cytotoxic effect consists in the experiments by using fibroblasts. The experiments with fibroblasts did not reveal any cytotoxic effect for both magnetized and non-magnetized scaffolds. The capture of fibroblastic cells (NIH/3T3) by various investigated scaffolds represented a logical follow-up of these tests confirming the ability of scaffolds to support cell viability and proving efficient cell adhesion especially for magnetic scaffolds. Our next step in the article was the investigations of MSCs as a precursor of osteogenic cell development toward osteoblastic lineage.

Summarizing, the efficiency of the scaffold seeding with magnetically labeled cell was evidently dependent on level of scaffold magnetization. Considering similar adhesive properties of fibroblasts and MSCs [82] the performed steps allowed to move from more standard investigation of the biocompatibility and adhesiveness towards investigation of scaffold influences on functional activity of cells, especially considering the relevant for practical medicine osteoinductive properties.

## 5. Conclusions

Magnetic scaffolds exemplify one of the most promising inputs of the nanotechnology to the area of tissue engineering and regeneration. Likewise, the 3D printing technology represents a powerful tool for the development of customized scaffolds to tackle several musculoskeletal pathologies characterized by bone loss with an enormous impact in the clinical framework. In our study, we combine these two innovative approaches and move on the development and upgrade of the efficient seeding of cells inside the scaffolds. The lack of toxic effects on both fibroblasts and MSC, and the intrinsic osteogenic potential of the newly developed PCL-HAp-1% SPION scaffolds, advance these scaffolds as promising candidates for bone tissue repair and regeneration. We believe the work will promote further investigations aiming at the deeper understanding of the biological mechanisms for MSCs and fibroblasts laden in PCL-HAp-1% SPION and unravel their potential clinical relevance.

## Figures and Tables

**Figure 1 polymers-13-03825-f001:**
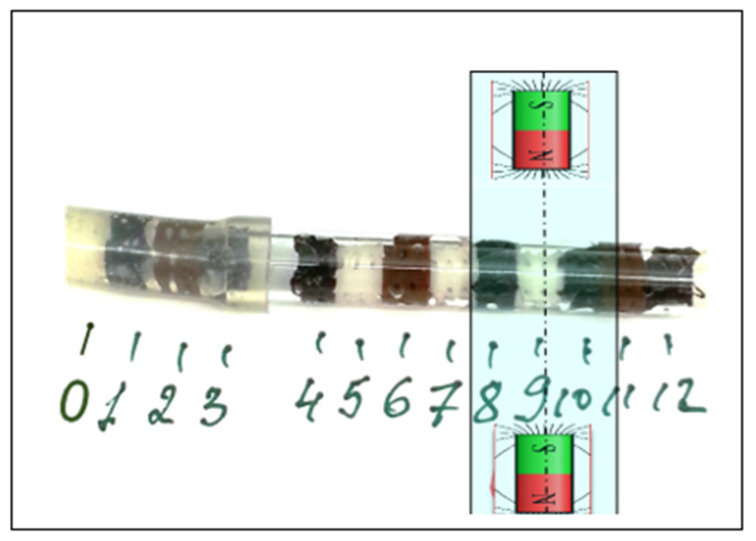
Experimental setup for cell seeding in the array of 13 scaffolds, numbered following their position. The scaffolds color corresponds to their magnetization: black for PCL-HAp-1%SPION, brown for PCL-HAp-0.5%SPION, and white for PCL-HAp.

**Figure 2 polymers-13-03825-f002:**
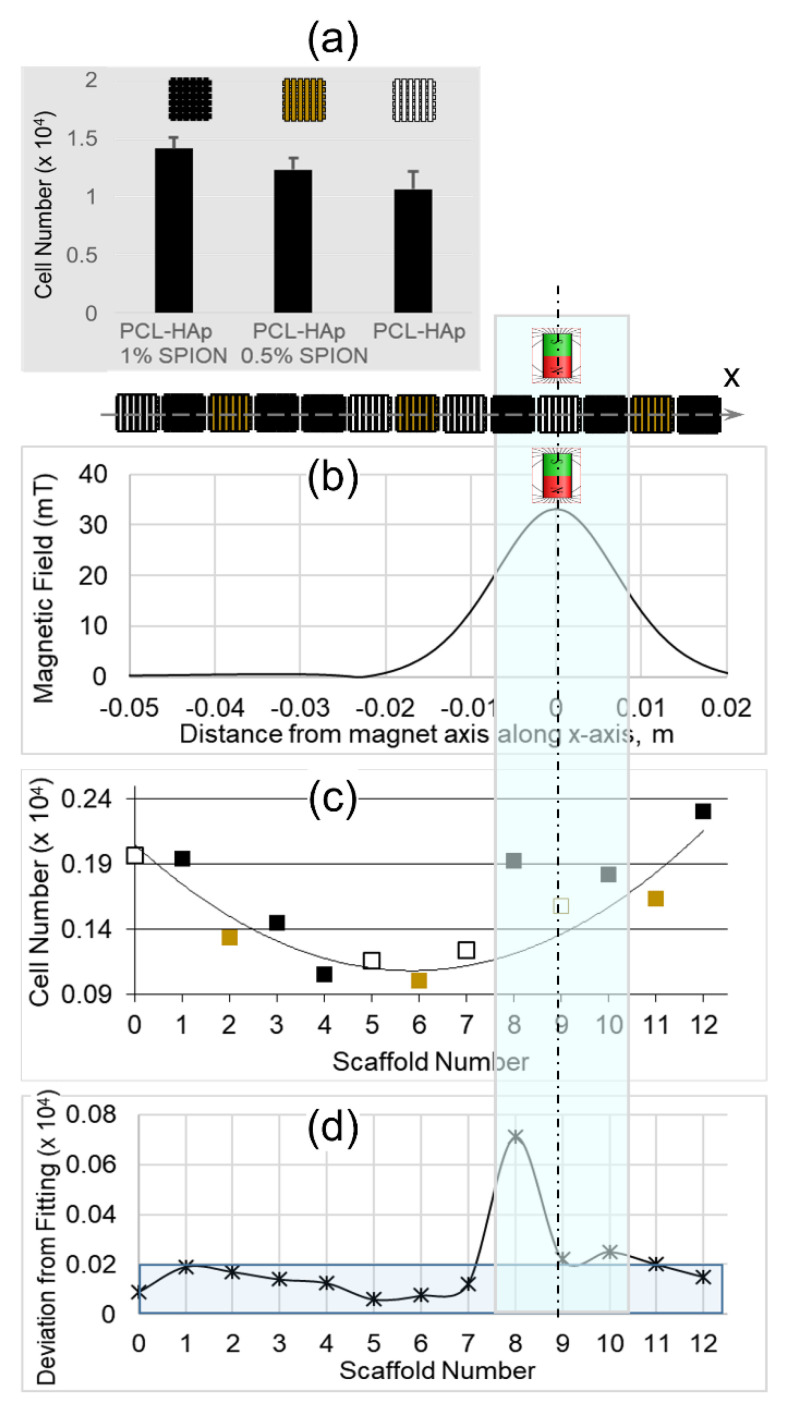
Cell seeding in the scaffolds with various levels of magnetization: (**a**) Number of the cells seeded in the single scaffolds; (**b**,**d**) Cell seeding in the array of 13 scaffolds (cyan highlighted area corresponds to maximal magnetic field intensity): (**b**) Magnetic field distribution along the scaffold centers (*x*-axes); (**c**) Cell amount seeded in the scaffolds and the fitting line the color of markers indicates the scaffold magnetization, while the scaffold numbers correspond to those indicated in Figure 1; (**d**) Deviation values of the loaded cell number from the fitting line.

**Figure 3 polymers-13-03825-f003:**
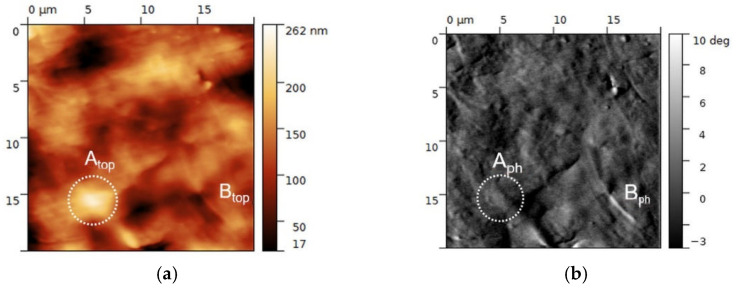
Representative 20 × 20 µm^2^ topographic (**a**) and magnetic phase (**b**) images of PCL-HAp-1%SPION. The correspondence between features A and B in terms of topographic (top) and magnetic phase (ph) signals is highlighted. A: ~4 µm large, no contrast in the magnetic phase, likely single HAp particle; B: little topographic contrast but remarkable magnetic contrast, likely the outcropping contour of a magnetic domain.

**Figure 4 polymers-13-03825-f004:**
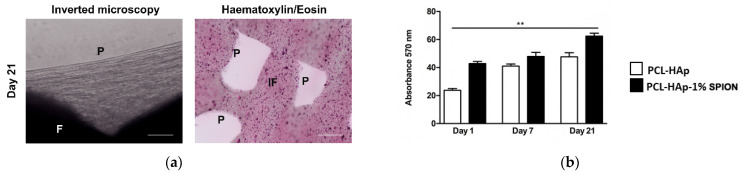
(**a**) Representative images from inverted light microscopy and hematoxylin/eosin staining of MSCs laden-3D PCL-HA-1% SPION at day 21, scale bar:100 µm; **P**: pores; **IF** inner filaments; **F** filaments. (**b**) Alamar blue assay of MSCs seeded onto PCL-HAPs (white bar) and PCL-HAp-1% SPION (black bar) at 1 and 21 days reported as absorbance at 570 nm. ** *p* < 0.01 MSCs seeded onto PCL-HAp-1% SPION at day 21 versus MSCs seeded onto PCL-HAp at day 1.

**Figure 5 polymers-13-03825-f005:**
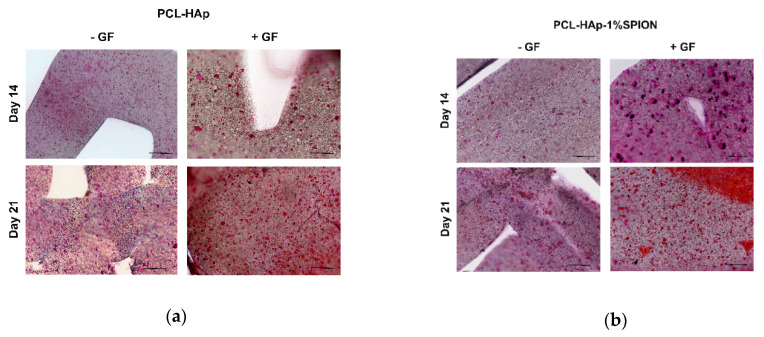
(**a**) Micrographs of PCL-HAp group, cultivated with and without GF (+/−GF), stained with Alizarin Red-S at 14 and 21 days. Scale bar: 100 µm. Red: mineralized areas. (**b**) Micrographs of PCL-HAp-1% SPION group, cultured with and without GF (+/−GF), stained with Alizarin Red-S on days 14 and 21. Scale bar: 100 µm. Red: mineralized areas.

**Figure 6 polymers-13-03825-f006:**
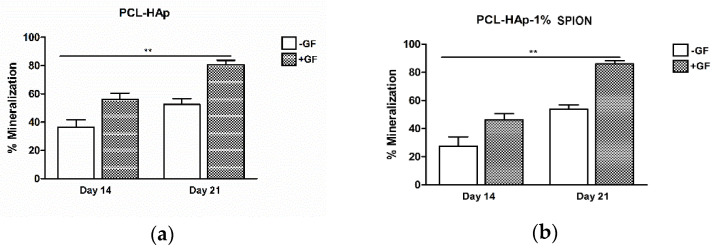
(**a**) Graphical representation of % mineralization of MSCs seeded onto PCL-HAp and cultured with (+) and without (−) growth factors (GF) at 14 and 21 days. Data are expressed as mean ± SD. Kruskall-Wallis and post hoc Dunn’s test were used for statistical analysis. ** *p* < 0.01 PCL-HAp cultured −GF on day 14 versus PCL-HAp + GF on day 21. (**b**) Graphical representation of % mineralization of MSCs seeded in PCL-HAp-1%SPION and cultured +/−GF on days 14 and 21. Data are expressed as mean ± SD. Kruskall-Wallis and post hoc Dunn’s test were used for statistical analysis. ** *p* < 0.01 PCL-HAp-1%SPION cultured −GF on day 14 versus PCL-HAp-1%SPION +GF on day 21.

## Data Availability

The data presented in this study are available on request from the corresponding author. We inserted some raw data in Appendix A.

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
