# Peer review of "Multifunctional 3D-Printed Magnetic Polycaprolactone/Hydroxyapatite Scaffolds for Bone Tissue Engineering"

_polymers, 2021, doi:10.3390/polym13213825_

Round 1

Reviewer 1 Report

In this article the author synthesized iron oxide functionalized hydroxyapatite  and  fabricate a scaffold  based on Fe nanoparticle HA/PCL, for bone tissue engineering. Although the provided information is not new but yet the article is well informative, and in this reviewer’s opinion add useful information to the body of the literature. I suggest the author to clearly define what is the new information here.  the article article to be considered for publication after revision detailed below:

please add the XRD spectra of the synthesized IOP HA with with without PCL

Why fibroblast cell lines was used when the target application is for bone? Osteoblast could be considered

The difference between samples with and without IOP  betterneed to be discussed also discussion required with other dopped polymers composites such as Zn doppped HA/PFF or PMMA, PCL etc that have been used for bone tissue engineering.

The porosity of the scaffolds need to be reported in addition to its biodegradation

Reviewer 2 Report

The paper is in field of bone tissue engineering- describing polycaprolactone (PCL)-based scaffolds added with hydroxyapatite (HAp) and superparamagnetic iron oxide nano- particles (SPION), able to drive on demand the necessary cells and other bioagents for high healing efficiency. The investigations are actual in would be useful for researchers working in polymer engineering field. The paper could be published after revision.

* AFM-MFM measurements confirmed a homogenous distribution of HAp and SPION throughout the surface. Would it be possible to find the distribution in all the 3D structure ?

* The magnetically assisted seeding of cells in the scaffold resulted most efficient for the 1% SPION concentration, providing good cell entrapment and adhesion rates. What would be comment of the authors, why the 1% SPION concentration is most efficient ? What range of the concentrations could be tested in the investigations ?

* In the introduction the authors should provide more information- what is already published in field of the bone tissue engineering by application of the superparamagnetic iron oxide nanoparticles ?

* Two different concentrations of SPION, 0.5 and 1% were selected, while HAp was included as 10 % of the total weight. It should be explained, why the concentrations were used ? Why other concentrations are not tested ?

* Maybe quality of the Figure 1 could be improved as for publication in scientific journal ?

* If the authors could clearly explain what are advantages of the developed 3D product as compared with those by employing polycaprolactone (PCL)-based scaffolds added with hydroxyapatite (HAp), but without superparamagnetic iron oxide nano-particles (SPION).

Round 2

Reviewer 2 Report

After the revision I would recommend for the Editor to consider acceptance of this paper for the journal.